# Back-Modality: Leveraging Modal Transformation for Data Augmentation

**Zhi Li**
Zhejiang University, China
zhili@zju.edu.cn

**Yifan Liu**
Zhejiang University, China
yifan.liu@zju.edu.cn

**Yin Zhang**[*]
Zhejiang University, China
zhangyin98@zju.edu.cn

## Abstract

We introduce Back-Modality, a novel data augmentation schema predicated on modal transformation. Data from an initial modality undergo a transformation to an intermediate modality, followed by a reverse transformation. This framework serves dual roles. On one hand, it operates as a general data augmentation strategy. On the other hand, it allows for other augmentation techniques, suitable for the intermediate modality, to enhance the initial modality. For instance, data augmentation methods applicable to pure text can be employed to augment images, thereby facilitating the cross-modality of data augmentation techniques. To validate the viability and efficacy of our framework, we proffer three instantiations of Back-Modality: back-captioning, back-imagination, and back-speech. Comprehensive evaluations across tasks such as image classification, sentiment classification, and textual entailment demonstrate that our methods consistently enhance performance under data-scarce circumstances.

## 1   Introduction

Neural network-based deep learning models are often prone to overfitting, resulting in a loss of generalization capability due to the limited size of training data. To mitigate this issue, data augmentation techniques are routinely employed to generate an augmented pool of training samples. The recent past has seen significant strides in the application of data augmentation within diverse fields such as speech [Ko et al., 2015, Park et al., 2019], computer vision [Simard et al., 1996, Shorten and Khoshgoftaar, 2019, Szegedy et al., 2014], and natural language processing [Sennrich et al., 2016, Wei and Zou, 2019]. Nevertheless, most of these data augmentation algorithms are tailored for a specific modality.

Concomitant with the swift advancement of cross-modal methodologies, various dual cross-modal models, such as text-to-image generation and image captioning, have demonstrated impressive performance. This progress catalyzes our development of a novel paradigm for data augmentation: Back-Modality, a comprehensive data augmentation framework predicated on modal transformation. Data in the initial modality are transformed to an intermediate modality and subsequently reversed back. Theoretically, the initial and intermediate modalities can represent any modality, including but not limited to text, image, audio, and video. Our framework retains its universality, contingent on the availability of corresponding dual cross-modal models to execute the modal transformation. Leveraging pretrained cross-modal models, our method is capable of generating a wide array of high-quality and diverse data. A comparison illustrating the breadth and quality of the generated data is provided in Figure 1.

Simultaneously, our framework enables the augmentation method intended for the intermediate modality to benefit the initial modality. For instance, data augmentation methods designed for pure

---

[*]Corresponding author: Yin Zhang.

37th Conference on Neural Information Processing Systems (NeurIPS 2023).

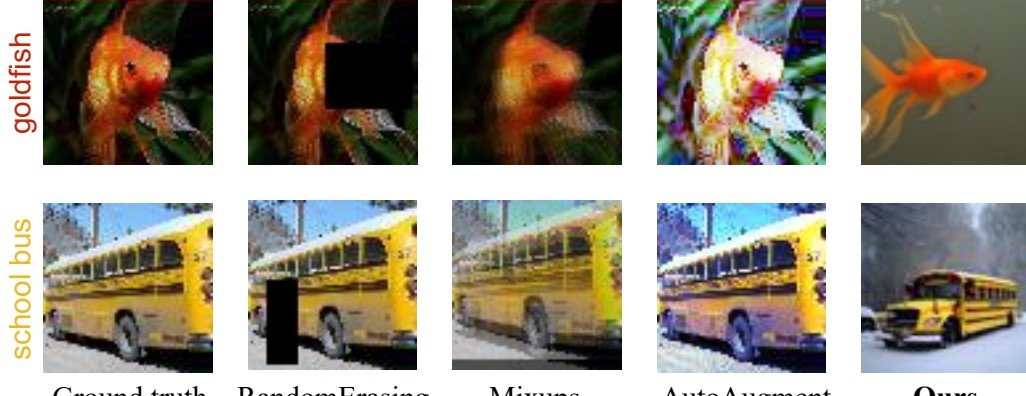

Figure 1: Comparative evaluation of images generated by our data augmentation method and other approaches. Capitalizing on the inherent capability of pretrained cross-modal models, our method excels in generating a broad range of high-quality and diverse images.

text can be leveraged to augment images, fostering cross-modality of data augmentation techniques. In essence, our framework facilitates partial decoupling of data augmentation methods from specific modalities. Moreover, our framework broadens the application landscape of cross-modal models. Given that our methodology obviates the need to access model weights or undergo further fine-tuning, it can be seen as an application variant of Cross-Modal-Models-as-a-Service (CMMaaS).

To substantiate the feasibility and efficacy of our framework, we present three instantiations of Back-Modality: back-captioning, back-imagination, and back-speech. For back-captioning, the initial modality is an image, and we utilize an image-caption model to generate captions for each image. Subsequently, data augmentation methods designed for text are employed to augment these captions. Finally, a text-to-image model is invoked to generate variant images, which are then used as augmented samples. For back-imagination and back-speech, the initial modality is text, and the intermediate modality is image and speech, respectively.

Systematic evaluations across various tasks including image classification, sentiment classification, and textual entailment demonstrate that our methods can consistently improve performance, particularly under data-scarce conditions. Upon further analysis and comprehensive case studies, it has been observed that the data generated via our method typically exhibit increased diversity.

Our principal contributions and discoveries can be summarized as follows:

- We introduce Back-Modality, a modality-agnostic data augmentation framework predicated on modal transformation. Within this framework, data in the initial modality undergo transformation to an intermediate modality and subsequently reverse back.

- Our framework enables the cross-modality of data augmentation methods. For instance, data augmentation techniques developed for pure text can be deployed to augment images.

- Our approach extends the application realms of cross-modal models. Our methodology eliminates the need to access model weights or conduct further fine-tuning, hence it can be perceived as a variant of the Cross-Modal-Models-as-a-Service (CMMaaS) application.

- Experiments on a variety of tasks and datasets substantiate that our methods can consistently enhance performance, particularly in data-scarce scenarios. Further analysis reveals that data generated by our method typically exhibits a higher degree of diversity.

## 2 Method

### 2.1 Framework

Back-Modality presents a comprehensive, modality-agnostic data augmentation framework predicated on modal transformation. As shown in Figure 2, $\mathcal{A}$ symbolizes the initial modality, and $\mathcal{B}$ signifies the

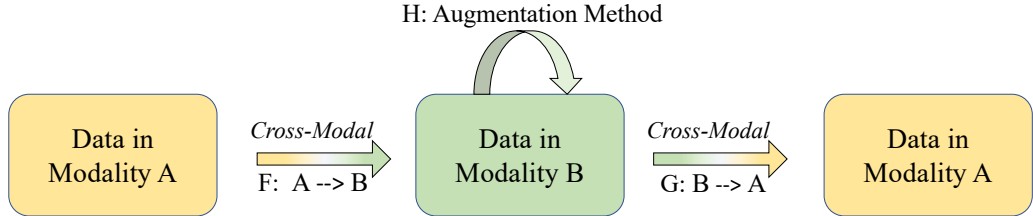

Figure 2: Back-Modality constitutes a universal data augmentation framework premised on modal transformation. $\mathcal{A}$ represents the initial modality, and $\mathcal{B}$ denotes the intermediate modality. The framework integrates two types of dual cross-modal tasks: $F : \mathcal{A} \mapsto \mathcal{B}$ and $G : \mathcal{B} \mapsto \mathcal{A}$. H signifies the data augmentation methods applicable to the intermediate modality $\mathcal{B}$.

intermediate modality. The framework incorporates two types of dual cross-modal tasks: $F : \mathcal{A} \mapsto \mathcal{B}$ and $G : \mathcal{B} \mapsto \mathcal{A}$. Here, $H$ denotes the augmentation methods applied to modality $\mathcal{B}$.

Let $X$ denote a sampled data point in the initial modality $\mathcal{A}$, and $X_{aug}$ represents the augmented samples of $X$. Our framework can be succinctly expressed as:

$$X_{aug} = G(H(F(X))) \tag{1}$$

Theoretically, $\mathcal{A}$ and $\mathcal{B}$ can represent any modality, such as text, image, audio, video, and so forth. Moreover, $\mathcal{A}$ and $\mathcal{B}$ can also be the same modality. For instance, back-translation augmentation methods can be perceived as a special case of our framework, where $\mathcal{A}$ and $\mathcal{B}$ are both text in different languages, and $H$ is omitted. Essentially, our framework retains its universality, contingent upon the availability of corresponding dual models.

Within our framework, $H$ can assist in augmenting data in modality $\mathcal{A}$ by augmenting data in the intermediate modality $\mathcal{B}$. This implies that the augmentation method suitable for a specific modality can be transferred to other modalities via our framework. More formally, let us consider a scenario where $F$ and $G$ are fixed, with $Y_{F,G}$ denoting the corresponding framework. Here, $H$ is treated as a variable. Consequently, the augmented samples $X_{aug}$ can be viewed as a function of $X$ and $H$.

$$X_{aug} = Y_{F,G}(H, X) \tag{2}$$

Given a specific $X$, varying $H$ can yield a diverse set of augmented samples $X_{aug}$. In essence, our framework facilitates partial decoupling of data augmentation methods from specific modalities.

## 2.2 Instantiations

In order to demonstrate the feasibility and efficacy of Back-Modality, we introduce three instantiations of this framework: back-captioning, back-imagination, and back-speech. These methods encompass three prevalent modalities: text, image, and speech.

**Back-captioning** In the proposed method, we define $\mathcal{A}$ as image modality and $\mathcal{B}$ as text modality. Here, $F$ represents the task of image captioning, while $G$ represents the task of generating images from text. $H$ corresponds to the process of textual data augmentation.

Back-captioning, as a concept, can be deconstructed into a trifecta of augmentation methodologies: multi-captioning, caption augmentation, and multi-imagination. It is observed that various observers usually generate captions for the same image that, while being semantically analogous, exhibit subtle variances in detail. This phenomenon mirrors the capability of most image captioning models to generate multiple unique captions for a single image, a process we refer to as multi-captioning. Given a caption sentence, several text augmentation techniques can be employed to produce multiple related sentences, a process we term as caption augmentation. For a given sentence, the images that different individuals visualize generally share similarities, yet present unique details. We refer to this phenomenon as multi-imagination. A diffusion-based model, employing various random seeds, can effectively simulate this process. Given an image captioning model that generates $l$ captions for each image, a caption augmentation process that generates $m$ sentences for each caption, and a text-to-image model that generates $n$ images for each sentence, the maximum total number of generated augmented sentences per original sentence amounts to $l * m * n$. This obviously represents

a highly efficient data augmentation approach. In practical implementation, we conduct uniform sampling at random on these augmented data to obtain the final augmentation dataset.

**Back-speech**     In this proposed approach, we designate modality $\mathcal{A}$ as text, while $\mathcal{B}$ pertains to the audio modality. Here, $F$ corresponds to the task of text-to-speech generation, and $G$ denotes the automatic speech recognition task. The text-to-speech model is designed to generate one audio file for each input sentence. Following this, audio data augmentation techniques, such as pitch shifting, denoted by $H$, are applied to these audio files to produce augmented sound files. Subsequently, the automatic speech recognition model translates these augmented sound files back into text, thereby generating augmented sentences.

The notion of back-speech can be elucidated by the following observation: the sentence articulated by an intermediary often bears slight deviations from the original sentence. This is due to the fact that different individuals may have distinct pronunciations for the same sentence, influenced by factors such as speech rate, pitch, or regional accents. To simulate these variations, we employ augmentation techniques within the audio modality. As a consequence, in the resultant output, some words may be substituted by their homophones. Given that most homonyms bear morphological similarities, with only minor differences at the subword level, this transformation can be viewed as a form of subword regularization facilitated by the audio modality.

**Back-imagination**     Back-imagination can be construed as the inverse operation of back-captioning. In this approach, we assign modality $\mathcal{A}$ to text, while $\mathcal{B}$ corresponds to the image modality. Here, $F$ represents the task of generating images from text, and $G$ signifies the image captioning task. Similarly, back-imagination amalgamates three types of augmentation techniques: multi-captioning, image augmentation, and multi-imagination.

| Method | Back-captioning | Back-speech | Back-imagination |
|--------|-----------------|-------------|------------------|
| F | OFA | fast-speech2 | stable-diffusion v2 |
| G | stable-diffusion v2 | wav2vec2 | OFA |
| H | augmentation with GPT | pitch shifting/time stretching | - |
| Task | image classifaction | sentiment classification | textual entailment |
| Dataset | Tiny ImageNet | SST-2 | TNCC |

Table 1: Statistics of configurations. "augmentation with GPT" means that we employ GPT model to augment captions. The symbol "-" indicates that we do not apply image augmentation in the process of back-imagination. Tiny ImageNet [Le and Yang, 2015] is an image classification dataset, SST-2 [Socher et al., 2013] is employed for sentiment classification, and TNCC, a textual entailment dataset, is based on Crisscrossed Captions [Parekh et al., 2020]. For further details, please refer to the appendix.

# 3   Experiments

In this section, we initially delineate the tasks, datasets, and models employed in our research. Subsequently, we assess the efficacy of back-captioning, back-speech and back-imagination in data-scarce scenarios, spanning a diverse range of tasks and datasets. Finally, we investigate the intrinsic mechanisms underlying the effectiveness of our proposed method from three perspectives: (1) conducting a quantitative analysis on the diversity and affinity of the generated samples, (2) performing ablation experiments to evaluate the contribution of individual components within the method, and (3) scrutinizing the differences between the original and augmented samples at an instance level and providing an in-depth analysis of the transformation effectuated by the augmentation process.

## 3.1   Baseline and Comparison

We use the performance of the base model, devoid of any data augmentation, as a baseline to evaluate the effectiveness of our augmentation method in enhancing model performance. For the image classification task, we employ the Resnet-18 model [He et al., 2015]. We use BERT [Devlin et al.,

2018] for sentiment classification and textual entailment tasks. To further substantiate the efficacy of our methods, we compare them against several other data augmentation strategies. For the image classification task, we consider four renowned methods: Random Erasing [Zhong et al., 2020], Autoaugment [Cubuk et al., 2019], Alignmixup [Venkataramanan et al., 2022], and Puzzle Mix [Kim et al., 2020]. For the sentiment classification and textual entailment tasks, we opt for the most popular task-agnostic data augmentation methods for comparison, as per Longpre et al. [2020], Yoo et al. [2021], including EDA [Wei and Zou, 2019] and Back-translation [Sennrich et al., 2016, Fadaee et al., 2017, Edunov et al., 2018, Ng et al., 2019]. We employ the large pre-trained EN-DE and DE-EN translation models [Ng et al., 2019, Yoo et al., 2021] for back-translation, which have a size order of magnitude comparable to the cross-model models used by Back-Modality methods, making them a robust baseline. In addition, we also consider some task-specific data augmentation methods for reference, including TMix [Chen et al., 2020], SSMix [Yoon et al., 2021], and Treemix [Zhang et al., 2022].

### 3.2 Datasets and Configurations

**Datasets** Tiny ImageNet [Le and Yang, 2015] and the Stanford Sentiment Treebank-2 (SST-2) [Socher et al., 2013] are widely used datasets. Given the scarcity of datasets for understanding natural language in visual scenes, we introduce a novel textual entailment dataset, named Textual Natural Contextual Classification (TNCC). And detailed descriptions can be found in the appendix.

**Data-scarce scenarios** To showcase our approach, we conduct experiments on artificially data-scarce tasks by sub-sampling the training set. For image classification, sentiment classification, and textual entailment tasks, we perform a class-balanced sub-sample on the training set and we denote the number of data samples for each class in the downsampled dataset as "shot," before any data augmentation is applied. We adhere to statistical rigor in our experiments by executing the sub-samples to 5 different data seeds. Concurrently, for each sub-sample, we train all models using 5 different random seeds. We report the statistical mean for all results. All experiments were conducted using PyTorch and executed on RTX 6000 GPUs and the Atlas computing cluster.

**Configurations** Table 1 provides a comprehensive view of the configurations used in our experiments. Unless explicitly stated otherwise, all pretrained models utilized in our research are obtained from the Huggingface Transformers library[2] [Wolf et al., 2019]. The code used to generate augmented samples and the TNCC dataset are both accessible[3]. The default augmentation size used in our studies, which includes both our method and the comparison method, is set to 5. As an illustration, in our experiments with Back-captioning, when we have $l = n = m = 2$, we create 8 augmentations for each image (2x2x2), and then, we randomly select 5 of these augmentations to form the final augmented dataset. For back-captioning, we utilize gpt-3.5-turbo [4], augmenting captions using the following prompt: "Maintain the nouns in the following sentence intact and generate semantically diverse sentences." For the process of back-imagination, we consciously exclude image augmentation in our experiments. One reason is that images produced via certain augmentation techniques, such as random erasing [Zhong et al., 2020] and cutout [DeVries and Taylor, 2017], often present a substantial challenge to image captioning models. On the other hand, the combination of multi-imagination and multi-captioning appears to be sufficient to yield satisfactory results.

**Detailed Strategies** In our experimental process, we observed that simply constructing a pipeline does not necessarily ensure the quality of the augmented data. Consequently, we have devised several strategies aimed at minimizing the production of low-quality or even inaccurate augmented data.

In the case of back-captioning, while the model possesses the capacity to generate highly relevant image descriptions, when deployed for specific tasks such as image classification, more nuanced categories are often required. However, the model sometimes falls short in providing these nuanced labels within the generated descriptions. For instance, when given an image of a 'Persian cat', the model tends to generate descriptions mentioning 'cat' rather than the more specific 'Persian cat'. Several image captioning models, such as the OFA [Wang et al., 2022] model, have the capability to generate image captions conditional on both a text prompt and an image. To address this issue, we

---

[2]https://github.com/huggingface/transformers
[3]https://github.com/zhilizju/Back-Modality
[4]https://platform.openai.com/docs/models/gpt-3-5

| Method | Shot | | | |
|---|---|---|---|---|
| | 1 | 3 | 7 | 10 |
| *Base-model* | 2.40 | 5.19 | 8.64 | 11.75 |
| *Random Erasing* | 2.61 | 5.77 | 9.11 | 12.59 |
| *Auto augment* | 3.41 | 6.35 | 9.84 | 13.23 |
| *Alignmixup* | 4.42 | 8.19 | 11.78 | 14.34 |
| *Puzzle Mix* | 4.48 | 10.26 | 12.52 | 15.66 |
| ***Back-captioning*** | **10.67** | **14.55** | **16.13** | **20.07** |

Table 2: The top-1 accuracy metric for back-captioning on the Tiny ImageNet dataset is displayed above. The overall accuracy is low, primarily because the model is trained from scratch using scarce data. "Shot" is used to indicate the quantity of images per label.

| Method | Shot | | | |
|---|---|---|---|---|
| | 1 | 2 | 5 | 10 |
| *Base-model* | 61.18 | 72.47 | 81.30 | 84.57 |
| *EDA* | 61.04 | 73.89 | 83.61 | 85.71 |
| *Back-translation* | 62.35 | 74.07 | 84.30 | 85.28 |
| *TMix* | 60.11 | 71.54 | 82.77 | 84.60 |
| *SSMix* | 62.32 | 74.47 | 82.89 | 85.32 |
| *Treemix* | 62.88 | 73.27 | 85.15 | 87.41 |
| ***Back-imagination*** | **69.20** | **79.62** | **88.41** | **89.14** |

Table 3: The accuracy of the back-imagination method on the TNCC dataset is indicated above. "Shot" signifies the quantity of sentence pairs corresponding to each label.

explicitly inject the image labels into the text prompts, which leads to the generation of descriptions that incorporate these finer-grained labels.

During the back-imagination process, there are occasions when the text-to-image model generates black and white images. This can lead to subsequent image captioning models generating sentences like "A black and white photo of," which are clearly inappropriate for augmentation samples. Consequently, if the back-imagination process yields a black and white image, we directly discard it and proceed with a resampling operation. As for back-speech, if the sentences generated have an edit distance from the original sentence that exceeds 20% of the entire sentence length, we discard those sentences as well.

## 3.3 Results

Tables 2, 3, and Table 4, respectively, display the statistical average accuracy values on the Tiny ImageNet, TNCC, and SST-2 datasets. We carried out hypothesis testing on the results across all tasks, with all p-values falling below 0.05. This suggests that our findings are statistically significant. Our back-captioning, back-imagination, and back-speech methods consistently outperform both the base model and other data augmentation methods. This indicates that in data-scarce scenarios, data augmentation based on modal transformation can offer significant performance improvements. In particular, the results of back-imagination demonstrate that our architecture alone, even without the inclusion of additional data augmentation methods (H), is capable of generating effective augmented data.

## 3.4 Ablation

Table 5 shows the ablation studies of back-captioning, back-imagination and back-speech. The individual components of our framework make significant contributions to the overall performance of the model. In particular, augmentation with GPT, pitch shifting and time stretching all improve the model performance, which strongly supports the notion that our framework effectively accomplishes the cross-modality of data augmentation approaches.

| Method | Shot | | | | |
|---|---|---|---|---|---|
| | 1 | 2 | 3 | 5 | 10 |
| *Base-model* | 51.46 | 52.31 | 54.42 | 57.88 | 61.10 |
| *EDA* | 50.35 | 52.17 | 54.87 | 58.32 | 61.72 |
| *Back-translation* | 51.30 | 52.09 | 55.84 | 57.69 | 61.94 |
| *TMix* | 49.71 | 51.59 | 54.18 | 57.62 | 61.31 |
| *SSMix* | 51.01 | 52.26 | 55.32 | 58.11 | 61.92 |
| *Treemix* | 51.35 | 52.18 | 55.81 | 58.73 | 62.37 |
| ***Back-speech*** | **52.13** | **52.78** | **56.11** | **59.03** | **63.21** |

Table 4: The accuracy of the back-speech on the SST-2 dataset is provided above."Shot" refers to the count of sentences associated with each label.

| Dataset | Method | Score |
|---|---|---|
| | *Base-model* | 11.75 |
| | *+Back-captioning* | 20.07 |
| *Tiny ImageNet* | *w/o multi-captioning* | 17.21 |
| | *w/o augmentation with GPT* | 18.49 |
| | *w/o multi-imagination* | 18.92 |
| | *Base-model* | 81.30 |
| | *+Back-imagination* | 88.41 |
| *TNCC* | *w/o multi-imagination* | 84.71 |
| | *w/o multi-captioning* | 86.52 |
| | *Base-model* | 57.88 |
| | *+Back-speech* | 59.03 |
| *STS2* | *w/o pitch shifting* | 58.45 |
| | *w/o time stretching* | 58.60 |

Table 5: Ablation studies. We report the accuracy of shot 10 for back-captioning, shot 5 for both back-imagination and back-speech. Multi-captioning: generate multiple captions for each image. Multi-imagination: generate multiple images for each sentence. Augmentation with GPT: augment captions with GPT model. Pitch shifting and time stretching are different augmentation methods used for speech.

## 3.5 Diversity and Affinity

Following the methodology outlined by Gontijo-Lopes et al. [2020], we further analyze our augmentation methods based on two key dimensions: Diversity and Affinity. Diversity is designed to quantify the notion that augmentations can prevent model overfitting by increasing the number of samples in the training set. As a metric of diversity, Gontijo-Lopes et al. [2020] employ the final training loss of a model trained with a given augmentation. The larger the loss function, the better the diversity of the augmented dataset. Affinity, on the other hand, quantifies how augmentation shifts data with respect to the decision boundary of the clean baseline model. Affinity is defined as the difference between the validation accuracy of a model trained on clean data and tested on clean data, and the accuracy of the same model when tested on an augmented validation set. Table 6 indicates that while ensuring comparable affinity metrics, our method consistently presents higher diversity metrics compared to other augmentation methods.

## 3.6 Case Study

Figure 1 showcases the images generated by our data augmentation method alongside other approaches. Upon further analysis, we have uncovered advantages in the back-captioning method that many traditional augmentation algorithms lack. As illustrated in Figure 3, we have listed several typical data augmentation patterns found in the back-captioning method:

1. **Luminance Modulation**: This involves altering the image's brightness by either decreasing or increasing it.

| Dataset | Method | Diversity | Affinity |
|---|---|---|---|
| Tiny imagenet | Base-model | 1.594 | 0 |
| | Random Erasing | 1.621 | -5.44 |
| | Auto augment | 1.615 | **-5.24** |
| | Back-captioning | **1.723** | -6.04 |
| TNCC | Base-model | 0.0279 | 0 |
| | EDA | 0.0343 | -1.14 |
| | Back-translation | 0.0301 | -0.91 |
| | Back-imagination | **0.0677** | **-0.82** |
| SST-2 | Base-model | 0.0089 | 0 |
| | EDA | 0.0126 | -7.57 |
| | Back-translation | 0.0104 | **-6.88** |
| | Back-speech | **0.0154** | -7.93 |

Table 6: Statistics of diversity and affinity. The former measures the variety of outcomes, with higher values indicating increased diversity and thus considered favorable. The latter quantifies how augmentation shifts data with respect to the decision boundary of the clean baseline model, where a value closer to 0 signifies greater affinity.

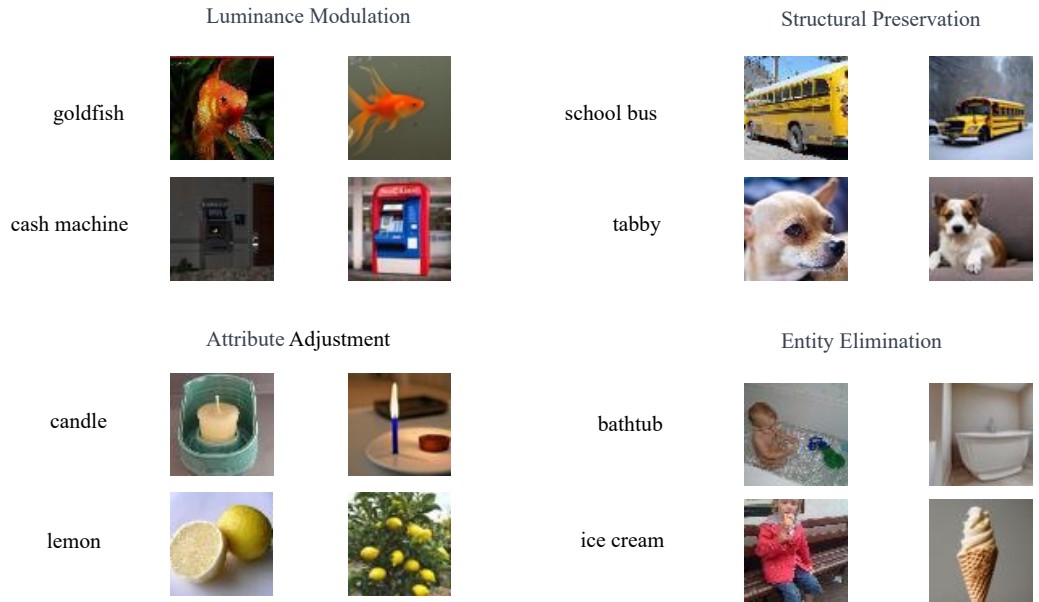

Figure 3: By comparing the original image with the augmented image using back-captioning method, we find typical data augmentation patterns: **Luminance Modulation**: Adjusting the image brightness either by decreasing or increasing it. **Structural Preservation**: Maintaining a high similarity of the target structure while altering the background. **Attribute Adjustment**: Modifying the target's attributes such as size, thickness, and quantity. **Entity Elimination**: Removing non-target entities.

2. **Structural Preservation**: This technique focuses on preserving the primary subject's structure while making changes to the background elements.

3. **Attribute Adjustment**: Adjustments are made to the target's attributes, such as size, thickness, and quantity.

4. **Entity Elimination**: This technique involves the removal of non-target entities from the image.

We also perform an extensive case study on back-imagination and back-speech. As depicted in Figure 4, we notice that augmented sentences, generated by different modal transformations, display unique characteristics. Further analysis reveals three notable advantages of our methods:

```
Some augmented cases of back-imagination:

(1)  Add/ Remove detail
    The black dog runs through the water .  ⟶ A black dog running through the water in a lake .
    A young man in a black and yellow jacket is gazing at something and smiling .  ⟶ A young man in a black and yellow jacket is smiling .
(2) Replace word by synonym/ hypernym/ hyponym
    A woman is throwing a frisbee on a beach .  ⟶ A woman is playing a frisbee on a beach .
    A man is standing in front of a skyscraper .  ⟶ A man standing in front of a building .
    Two children sit on a small seesaw in the sand .  ⟶ Two boys sit on a small seesaw in the sand .
(3) Reverse  the order
    A boy strolls by a pond in a park .  ⟶ A  boy walks in a park near a pond.
(4) Change /blur the quantity
    Three people are walking on a path in a meadow .  ⟶ A group of people are walking on a path in a meadow .
(5) Tense variation
    Two children sit on a small seesaw in the sand .  ⟶ Two children  is sitting on a small seesaw in the sand .
...

Some augmented cases of back-speech:

 (1) Split /merge the words
    That 's far too tragic to merit such superficial treatment .  ⟶ That is far too tragic to merit such superficial treatment .
    Remains utterly satisfied to remain the same throughout .  ⟶ Remain sutterly satisfied to remain the same throughout .
(2). Remove word
    He hoped there would be stew for dinner, turnips and carrots .  ⟶ He hoped there would be stew for dinner, turnips .
(3)  Replace word by homonym
    This is a shot of cannery row in 1932 .  ⟶ This is a shot of cannery row in nineteen thirty-two .
    Lend some dignity to a dumb story .  ⟶ Lend some dignity to a dome story .
    The year 's best and most unpredictable comedy .  ⟶ The nears best and most unpredictable comedy .
(4)  Change punctuation
    So, we have this ability as well .  ⟶ So we have this ability as well .
...
```

Figure 4: By comparing the original sentence with the augmented sentence, we can summarize several types from the perspective of linguistics. It is obvious that the linguistic transformations of our methods are diverse.

1. Leveraging cross-modal models, our back-imagination augmented sentences demonstrate exceptional fluency, readability, and minimal spelling or grammar errors. Correspondingly, the generated images consistently maintain a high level of quality.

2. Our approach introduces greater diversity in both augmented sentences and images. Notably, augmentation types such as adding details are challenging to achieve using popular text augmentation methods like EDA and back-translation.

3. Our method ensures better semantic consistency. For instance, while both EDA and our method involve deletion operations, our approach selectively removes details. But EDA performs random deletion, which increases the likelihood of altering the original semantics. A thorough analysis of back-speech augmentation reveals that certain words in augmented sentences are substituted with similar-sounding words, often sharing similar but not identical subwords. Additionally, the augmentation process involves splitting and merging certain words. Consequently, the augmentation of back-speech can be considered as a form of regularization akin to techniques such as BPE-Dropout [Provilkov et al., 2020] and SwitchOut [Wang et al., 2018].

# 4   Related Work

## 4.1   Data Augmentation

Data augmentation is widely utilized in various domains to address limited data scenarios, and it has achieved significant success. In computer vision, basic image manipulations such as translation and rotation can generate new samples [Ronneberger et al., 2015, Krizhevsky et al., 2017]. Techniques like random erasing [Zhong et al., 2020] and cutting [Devries and Taylor, 2017] aid in improving generalizability by occluding images. Pitch shifting and time stretching are popular methods in speech processing [Salamon and Bello, 2016, Moreno-Barea et al., 2018]. Text augmentation methods typically involve word-level text editing operations and sentence-level text generation.For word-level text editing operations, word substitution is commonly employed, including techniques such as

synonym replacement [Zhang et al., 2015], KNN replacement [Wang and Yang, 2015], Uniform and TF-IDF replacement [Xie et al., 2020], Bi-RNN Contextual replacement [Kobayashi, 2018], CBERT [Wu et al., 2019], and LAMBADA [Anaby-Tavor et al., 2019]. EDA [Wei and Zou, 2019] comprehensively explores text editing operations, including synonym replacement, insertion, swap, and deletion, for data augmentation. The popular text augmentation method based on generation is back-translation [Sennrich et al., 2016, Fadaee et al., 2017, Edunov et al., 2018]. It is evident that previous data augmentation methods have been primarily tailored for specific modalities.

### 4.2 Dual Cross-modal Tasks and Models

In recent years, large-scale pretrained foundation models have achieved remarkable success across a spectrum of tasks. Contemporary research [Zeng et al., 2022, Wu et al., 2023a, Shen et al., 2023] suggests the amalgamation of multiple large pretrained models for the execution of novel downstream multimodal tasks. However, our approach diverges from the aforementioned studies, as we employ dual cross-modal models to devise a fresh data augmentation paradigm. There exists a plenitude of dual cross-modal tasks and pretrained models, such as text-to-image [Ho et al., 2020, Rombach et al., 2022], image captioning [Wang et al., 2022, Li et al., 2022], text-to-video [Singer et al., 2022, Hong et al., 2022], video captioning [Yan et al., 2022, Yamazaki et al., 2022], text-to-speech [Ren et al., 2020], and automatic speech recognition [Baevski et al., 2020]. Dual Learning [He et al., 2016, Xia et al., 2017, 2018] protocols train the models of two dual tasks in tandem, while consciously leveraging the probabilistic correlation between them to regularize the training process. In contrast, our methodology primarily centers on the usage of the large pretrained models of two dual tasks for data augmentation, circumventing additional training.

## 5 Limitations

Compared to existing data augmentation techniques, Back-Modality typically requires additional computational resources and inference time due to its reliance on large pretrained cross-modal models. However, as advancements continue in the cross-modal field and efficient machine learning methodologies are further employed, we anticipate the development of more efficient techniques. As such, this limitation is expected to be progressively alleviated. Furthermore, while our framework is inherently versatile, specific instances may require the design of corresponding strategies akin to those discussed in Section 3.2, which ensure label invariance in the generated samples and maximize their affinity with the original data.

## 6 Conclusion

In this work, we introduce a novel data augmentation framework, predicated on modal transformation, and confirm its feasibility and effectiveness. We posit that as the field of cross-modal learning continues to evolve, modal transformation can present a fresh avenue for developing data augmentation techniques. This emerging field presents a plethora of intriguing research questions that merit further exploration. For instance, a comprehensive evaluation of our method across a broader spectrum of tasks and modalities would be insightful. In addition, investigating the impact of fine-tuning the cross-modal models on domain-specific datasets on the quality of the generated data could yield meaningful insights.

## Acknowledgments

This work was supported by the NSFC project (No. 62072399), Zhejiang Provincial Natural Science Foundation of China under Grant No. LZ23F020009, China Research Centre on Data and Knowledge for Engineering Sciences and Technology, MoE Engineering Research Center of Digital Library, and the Fundamental Research Funds for the Central Universities. Thanks to the Hangzhou AI Computing Center for providing the Atlas computing cluster. We also express our sincere gratitude to anonymous reviewers for their invaluable feedback and constructive comments, which significantly contributed to the improvement of this paper.

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

## Appendix

## 7  Back-imagination and Back-speech

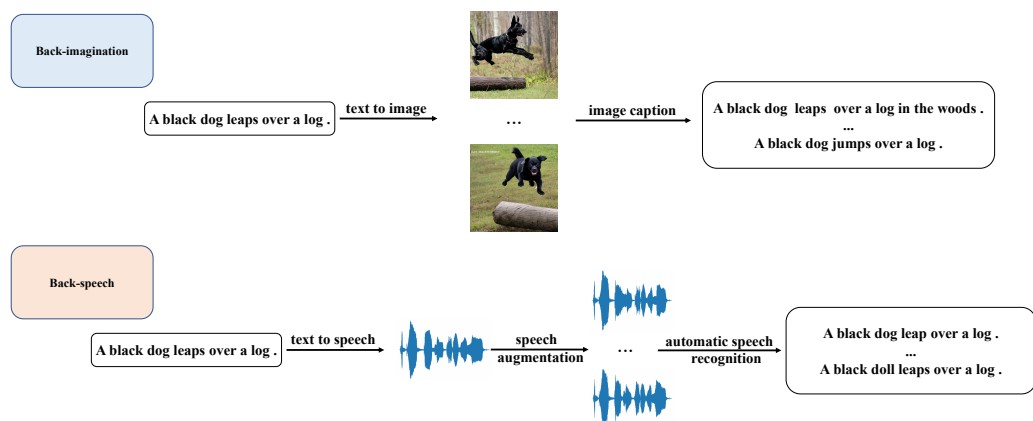

Figure 5: The illustrative examples for two proposed techniques: Back-imagination and Back-speech.

As shown in Figure 5, we present illustrative examples to facilitate a better understanding of two proposed techniques: Back-imagination and Back-speech.

## 8  Datasets

Tiny ImageNet [Le and Yang, 2015] serves as a compact version of the comprehensive ImageNet dataset. It comprises 100,000 images spanning 200 classes, with 500 images per class, and these images are downsized to 64×64 pixels. Each class is furnished with 500 training images, 50 validation images, and 50 test images.

The Stanford Sentiment Treebank-2 (SST-2) [Socher et al., 2013] is a sentiment classification dataset populated with movie reviews gathered from Rotten Tomatoes, paired with their corresponding binary labels. The dataset is partitioned into training, validation, and testing sets, comprising 67,349, 872, and 1,821 instances, respectively.

Given the scarcity of datasets for understanding natural language in visual scenes, we introduce a novel textual entailment dataset, named Textual Natural Contextual Classification (TNCC). This dataset is formulated on the foundation of Crisscrossed Captions [Parekh et al., 2020], an image captioning dataset supplied with human-rated semantic similarity ratings on a continuous scale from 0 to 5. We tailor the dataset to suit a binary classification task. Specifically, sentence pairs with annotation scores exceeding 4 are categorized as positive (entailment), whereas pairs with scores less than 1 are marked as negative (non-entailment). The TNCC dataset is partitioned into training, validation, and testing sets, containing 3,600, 1,200, and 1,560 instances, respectively. This dataset will be made available alongside our source codes.

## 9  Configurations

In this work, we employ a uniform experimental configuration for both textual entailment and sentiment classification tasks. We adopt BERT-BASE [Devlin et al., 2018], a model pretrained using Masked Language Modeling (MLM), as our primary experiment subject. For each individual downstream classification experiment, the classification model is initialized with the pretrained parameters from the BERT-BASE model. The classifier component, comprising of two fully connected layers that deduce class labels from the output embeddings generated by the transformer architectures, is randomly initialized. During the training phase, we leverage the Adam optimization algorithm with a learning rate set at $5e - 5$, the first and second momentum terms, $\beta_1$ and $\beta_2$, are respectively set to 0.9 and 0.999. Additionally, we introduce an $L_2$ weight decay of 0.01 to the model. We select a batch

size of 2 for all trials. We save model checkpoints during training and ultimately employ the best checkpoint—determined based on performance on the validation dataset—for testing. The results are presented as classification accuracies on both datasets under investigation.

For the image classification task, we employ the ResNet18 [He et al., 2015] model, which is considered more suitable for small datasets. We initialize all learnable layer parameters randomly. During the training process, we employ the SGD optimizer with a learning rate of 0.1, momentum of 0.9, and a weight decay of 0.0001.

## 10 Human Evaluation on Augmented Samples

In response to your suggestion, we conducted a human evaluation on the sampled augmented data. The results of the evaluation are as follows:

**For the images generated using the back-captioning method:**

- Label Invariance Score: 99.2%

**For the sentences generated using the back-imagination method:**

- Semantic Consistency Score: 98.8%

These high scores indicate that both methods performed exceptionally well in their respective evaluations. The results affirm that Back-Modality preserves the essential characteristics of the original data while introducing diversity, further validating our approach.

## 11 More Choices of Cross-Modal Generation Models

In our paper, for the Back-captioning with a 10-shot setting, we primarily used the OFA-large model, which yielded a top-1 accuracy of 20.07%. To assess the impact of different model sizes on the outcomes, we also conducted experiments with OFA-huge under the same conditions. The results showed a significant improvement, with the top-1 accuracy reaching 22.12%.

## 12 Cost of Obtaining the Augmented Samples

| Method | Additional Computational Time |
|---|---|
| RandErasing | 4 m 55 s |
| Puzzle Mix | 1 h 29 m 25 s |
| Alignmixup | 1 h 59 m 45 s |
| Back-captioning (our method) | 11 h 35 m |
| Auto augment | About 49 h |

Table 7: The additional computational overhead of various augmentation methods compared to the base model.

Table7 illustrates the additional computational overhead of various augmentation methods compared to the base model on RTX A6000. The primary cost of our method is related to generating images with the diffusion model, and the primary overhead of auto augment is associated with learning augmentation policies. In terms of text augmentation, on the textual entailment task, our method, back-imagination, took 4 hours 13 m 45 s, while the back-translation method took 5 h 38 m 12 s. On the sentiment analysis task, back-speech took 35 m 27 s, whereas the back-translation method took 5 h 22 m 4 s.

## 13 Discussion About Real-World Applicability

While current multi-modal and cross-modal models have achieved impressive results and continue to rapidly advance, it is worth noting that not all domains currently have easy access to open-source

cross-modal models. This limitation can, to some extent, restrict the effectiveness of our method in real-world applications. However, recent research has increasingly focused on the adaptability of diffusion-based cross-modal models in domains with limited data. This research encompasses areas such as few-shot [Giannone et al., 2022], one-shot [Wu et al., 2023b], zero-shot [Li et al., 2023], domain adaptation [Kim et al., 2023], and unsupervised domain adaptation [Benigmim et al., 2023]. These research directions will further expand the real-world applicability boundaries of our method.

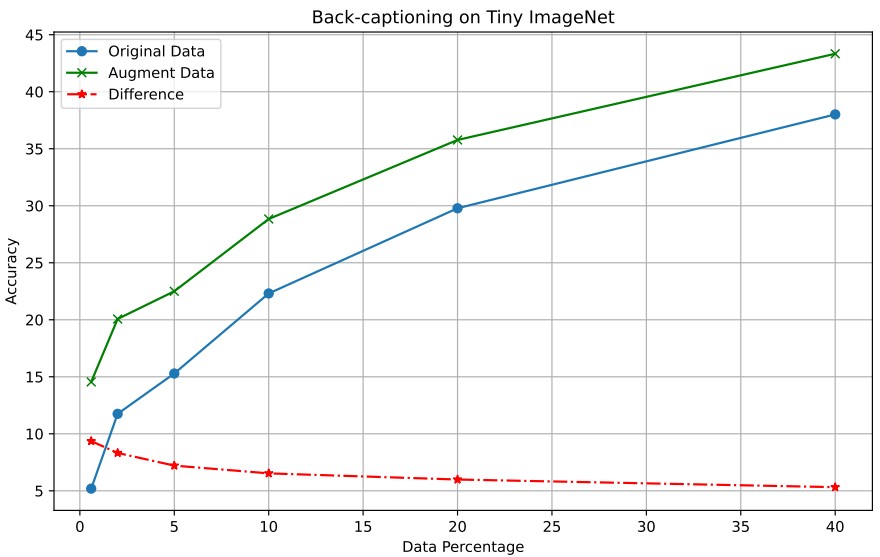

Figure 6: Effect of increasing original data.

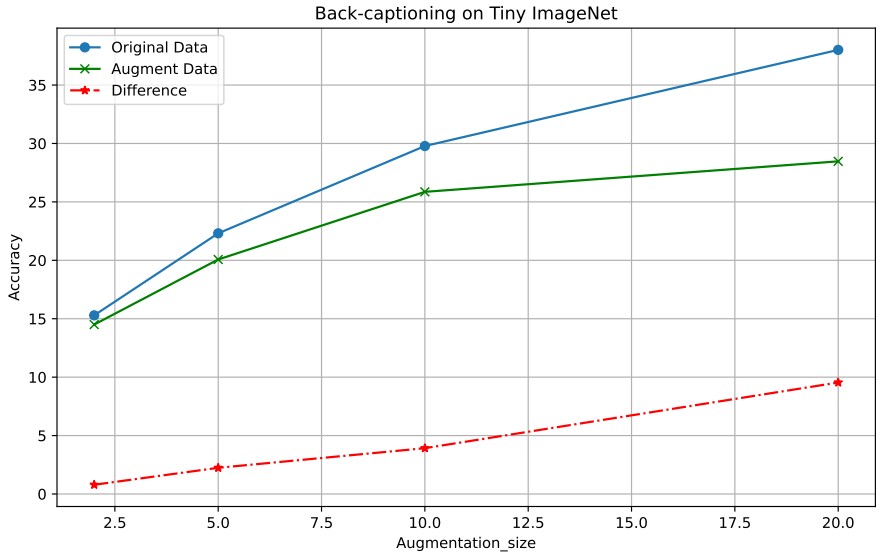

Figure 7: Effect of increasing augmentation multiple.

# 14 Proportion and Size

We conducted an analysis of the proportion of augmented data and the augmentation size by sampling a subset of the data. Figure 6 illustrates the variation in the model's performance when increasing the volume of original data while keeping the augmentation multiple at 5. From the curve in this figure, it is evident that as the volume of original training data increases, augmented data continues to provide benefits. However, these benefits exhibit diminishing returns as more original data is added. Figure 7 addresses the aspect of extending data generation. It demonstrates that as the augmentation size increases, the model's performance improves. However, after a certain point, the gains tend to plateau. We believe the primary reason for the phenomena observed in these two figures is that the diversity introduced by augmented data leads to performance gains. However, this diversity may not match the affinity of the original data, which can result in diminishing returns.

