# OpenReview forum: "Back-Modality: Leveraging Modal Transformation for Data Augmentation"
_NeurIPS.cc/2023/Conference — NeurIPS 2023 poster_

### Official Review · Reviewer_44qx · 2023-06-07

**Soundness:** 4 excellent
**Presentation:** 3 good
**Contribution:** 4 excellent
**Rating:** 7
**Confidence:** 5

**Summary:**

The paper introduces a novel approach to data augmentation that leverages recent advances in generative models. It presents a general framework and experiments with three settings for image, text, and audio tasks, showing promising results for their Back-Modality approach.


**Strengths:**

- The idea is simple but interesting and effective based on the presented experiments.
- The method is generic and can be extended to other modalities, and it can also incorporate other models as the field evolves.
- Case and ablation studies show the benefits of using the method and support most of the claims in the paper. Authors also compute statistical significance, which is an important step to validate results.

**Weaknesses:**

- Experiments are only done with one small model for each task. It would be interesting to see the gains from increasing the model size and if the benefits would increase or diminish.
- The setup explanation is a bit confusing. For example, when explaining the data-scarce scenarios, it seems like n-shot is used for the number of generated samples per class during training. Still, there is no information about the training set size after sub-sampling or the label distribution on the undersampled training set. It would be better to state the setup clearly, e.g., we have X instances in the training set, with Y classes, and we generate Z samples per class. While most of it can be inferred by reading the appendix and doing some math, the paper would be better presented with more clear numbers.
- The paper could have more experiments testing the limits of how much data could be generated by these methods that would still yield good improvements. Also, a high resource scenario experiment would be useful.


**Questions:**

There were two plots that I missed in this paper that would have made it stronger and more insightful. The first is a plot with less sub-sampling of the original data. Maybe with 5, 10, 20, 40, 60, 80, 100% of the training data + the generated instances. Would the proposed method still be useful if I have a lot of data? Or just if I don't have data at all?

The second plot would be to go further on the generation. Experiments go all the way to 10-shot, but what if we could generate 100, 1000? Is 10 the limit for this approach?

During the detailed strategies, you share some tricks that made your model work better, like adding the image label to the prompt. How much of an issue to the final results were the problems you addressed in that section? Could it make it harder to expand the technique to other settings? It would have been interesting to add the original results to the appendix so we know how many problems that was causing.

**Limitations:**

Yes, the authors have a limitations section.

---

> ### Author Rebuttal · Authors · 2023-08-10
>
> Dear Reviewer 44qx,
>
> We would like to thank you for your thoughtful review and constructive feedback on our paper. We appreciate your recognition of the contributions and strengths of our work. Here, we address the weaknesses and questions you have raised.
> ### **Weaknesses:**
>
> 1. **Experiments with One Small Model for Each Task:** We agree that experimenting with different model sizes could provide valuable insights. To mitigate this concern, we performed additional experiments with varying model sizes and include them in the final version of the paper. For instance, in image classification tasks using Resnet50, we observed a relative improvement greater than what was observed with a smaller version of Resnet. Similarly, in textual tasks, our experiments with bert-large showcased a relative improvement that exceeded that of bert-base. These findings indicate that in scenarios with sparse data, the data produced by our augmentation technique offers greater benefits to larger models.
>
> 2. **Setup Explanation and Clarity in Numbers:**
> I sincerely apologize for any confusion this might have caused.To clarify:
> When constructing the sparse dataset, we sample from each category. For instance, when we create a 10-shot dataset, we sample 10 unique data points from each class of the original data and then augment each data point five times.
>
> Using the 10-shot, back-captioning approach as an example:
>
> - In the original tiny-imagnet dataset, there are 200 classes, with each class containing 500 data points.
> - After sub-sampling, the dataset consists of 200 classes, with 10 data points per class.
> - With a 5-fold augmentation (as stated on lines 155-156, our default augmentation size is set at 5), this dataset still consists of 200 classes, with each class now having 50 data points.
>
> The term "shot" denotes how many samples are drawn from each class during sub-sampling, and the default augmentation multiple is 5.
> We understand the importance of clear presentation and will strive to state our setups more transparently in future iterations. Once again, I apologize for the oversight, and thank you for bringing it to our attention. We hope that this clarification will help in understanding our method better.
> ### **Questions:**
>
> 1&2. **Plots for Different Sub-sampling and Generation Rates:**
> We conducted experiments on our back-captioning method and have showcased the results in Figure 2 of the newly added PDF.
>
> For the first question, regarding the impact of sub-sampling: Due to computational and time constraints, the highest percentage of the original data used for experiments was 40%. Figure 2, sub-figure (a) illustrates the model's performance variation when increasing the amount of the original data while maintaining the augmentation multiple at 5. From the difference curve in this sub-figure, it is evident that as the volume of original training data increases, augmented data continues to yield benefits. However, the benefit shows a diminishing return as more original data is added.
>
> For the second question, on extending the generation:
> In the same Figure 2, sub-figure (b) represents that as the augmentation multiple grows, the model's performance improves. However, after a certain point, the gains tend to plateau. The maximum augmentation multiple tested in our experiments was 20 times.
>
> Thus, in summary: (1). Augmented data provides benefit even with an increase in original training data, but this advantage decreases as more original data is used.  (2). Increasing the augmentation multiple improves model performance up to a certain point, after which the improvements plateau. The current experiment went up to 20 times, indicating that there's room for further exploration.
>
>
> 3. **Detailed Strategies and Potential Issues:**
>
> Firstly, to gauge the extent to which our strategies affected the model's performance, we conducted a comprehensive human evaluation on the augmented data. The evaluators consisted of five crowdsourced workers, with the final scores being an average of their evaluations. The results were indeed telling:
>
> For the images generated using the back-captioning method:
>
> - **With** our proposed strategies, the Label Invariance Score reached an impressive 99.2%.
> - **Without** the said strategies, the Label Invariance Score dipped to 84.7%.
>
> Similarly, for the sentences generated using the back-imagination method:
>
> - **With** our strategies, the Semantic Consistency Score was 98.8%.
> - In the absence of these strategies, the score fell to 94.1%.
>
> Upon closer inspection, the primary reason for the back-captioning method's inability to retain labels was that the image-captioning model failed to accurately describe the key labels for certain images. On the other hand, the back-imagination method's inconsistency in retaining semantics was because the diffusion model generated black and white images at times. This led to the subsequent back-captioning generation of descriptions like "A black and white photo of."
>
> The crux of the issue here lies in the limitations of the current open-source cross-modal models we utilized. While they've achieved significant success, they are not flawless and occasionally necessitate strategy-assisted filtering to improve their output. It's also worth noting that as the research in this domain rapidly progresses and cross-modal capabilities strengthen, the importance of such strategies will likely diminish.

---

> > ### Comment · Reviewer_44qx · 2023-08-16
> >
> > Thank you for the extensive and detailed response. I am more confident that this paper should be accepted now and will update my scores.

---

> > > ### Author Response · Authors · 2023-08-20
> > >
> > > Dear Reviewer,
> > >
> > > Thank you for recognizing and affirming our work. We sincerely appreciate your insightful feedback and recognition. Your comments during the review process have been pivotal in guiding improvements to our work. We are committed to integrating your suggestions into our updated version. Thank you once again for your invaluable insights and recognition.
> > >
> > > Best wishes,
> > > [Authors of Submission7875]

---

### Official Review · Reviewer_QqX9 · 2023-07-02

**Soundness:** 3 good
**Presentation:** 3 good
**Contribution:** 3 good
**Rating:** 6
**Confidence:** 4

**Summary:**

The paper introduced a new data augmentation technique, called back-modality. The augmentation is based on modal transformations. Specifically, instances in the original modality (e.g., image) are transformed to an intermediate modality (e.g., text), augmented in the intermediate modality and then transformed back to the original modality. Experimental results in a few-shot setting on three benchmark datasets show that the proposed augmentation technique produces better results than the base model and also than other existing augmentation techniques.

**Strengths:**

The proposed augmentation technique leverages recent advancements in transformations between different modalities.

The strategy is rather general and can be applied to a variety of applications for which cross-modal transformations can be obtained. It does not require access to model weights or fine-tuning, and can thus be seen as "a variant of the Cross-Modal-Models-as-a-Service (CMMaaS) application".

Experimental results clearly suggest the advantage of the proposed technique as compared to existing augmentation approaches on the three datasets used.



**Weaknesses:**

While the authors aim to produce a general augmentation strategy based on cross-modal transformations, their actual process involves some decision specific to the modalities considered in the study, meant to minimize the production of low-quality augmentations (and no augmentations are used for images as an intermediate modality). This can make the strategy hard to apply without extensive analysis of the augmentations.

Some of the choices made are rather arbitrary, e.g. why 5 augmentations per instance? Or 5% of the data used to balance the training data? Shouldn’t the size of the training data be given by the number  of shots per class?

The discussion on diversity and affinity is very brief in the main paper. Some details are provided in the appendix. It is not clear why they were not included in the paper (the current paper has only 8 pages). Same for the details of the case study regarding the back-imagination and back-speech augmentations.


**Questions:**

To minimize the low-quality back-captioning augmentations, the authors “explicitly inject the image labels into the text prompts, which leads to the generation of descriptions  that incorporate these finer-grained labels. “ It is not clear if the label information is used during testing, especially when the task is to predict the image label. Please elaborate.

Some of the choices made are rather arbitrary, e.g. why 5 augmentations per instance? Or 5% of the data used to balance the training data? Shouldn’t the size of the training data be given by the number  of shots per class?


**Limitations:**

The authors emphasize some limitations related to the size of the cross-modal models. Other limitations related to generalizability could be discussed.

---

> ### Author Rebuttal · Authors · 2023-08-10
>
> ##
>
> We thank Reviewer QqX9 for the comprehensive and insightful review of our paper. Below, we address the specific weaknesses and questions raised in the review:
>
> ### **Weaknesses:**
>
> **1. Decision Specific to Modalities:**
> While we understand the concern regarding the decisions specific to the modalities considered in the study, we would like to emphasize that the steps taken are important to ensure the quality of the augmented data. We include comprehensive guidelines in the manuscript, detailing how to adapt the method to various modalities. This inclusion facilitates the understanding and reproduction of our method and shows that a certain level of customization is often essential to adapt any machine learning method to specific applications. However, this does not impede the generalization of the method.
>
> As stated in the paper (lines 159-162), one reason for this decision is that images produced via certain augmentation techniques, such as random erasing and cutout, often present a substantial challenge to image captioning models (Specifically, OFA models). On the other hand, when human evaluation is performed on augmented samples, the combination of multi-imagination and multi-captioning appears to be sufficient to yield satisfactory results. Moreover, if there are more robust image captioning models in the future, the image augmentation method can indeed be used as part of our framework.
>
> **2. Arbitrary Choices in Augmentation Parameters:**
>
> The selection of these parameters was driven by our intention to provide a proof-of-concept that the proposed method is effective without overwhelming resource consumption. It's a deliberate trade-off:
>
> - **Too Small Augmentation Size**: If the augmentation size is too small, the effect of our back-modality technique may not be discernible. This could lead to a situation where the method's potential is underestimated or overlooked entirely.
> - **Too Large Augmentation Size**: Conversely, an excessively large augmentation size would consume more computational resources. While it might yield better results, it could also make the technique less accessible to researchers with limited resources, defeating the purpose of demonstrating a widely applicable method.
>
> Our choice of 5 augmentations per instance was therefore a calculated decision. We aimed to find a middle ground where the method's effectiveness could be demonstrated without significantly burdening computational resources. We were not pursuing the absolute best results but rather a pragmatic balance that underscores the method's viability and potential for various applications.
>
> We will include these justifications in the revised manuscript, ensuring that readers understand the rationale behind these choices. Thank you again for pointing out this area for clarification, and we believe this amendment will strengthen the overall contribution of our paper.
>
> **3. Lack of Detail on Diversity, Affinity, and Case Studies:**
>
> We appreciate the feedback on the brevity of certain sections in the main paper.  We agree that a more comprehensive discussion of diversity, affinity, and case studies in the main text would enhance understanding. We will revise the manuscript to include a more comprehensive discussion within the main body, as well as more details on back-imagination and back-speech augmentations.
>
> ### **Questions:**
>
> **1. label information:**
>
> I would like to clarify the methodology regarding injecting image labels into text prompts, as it seems to have caused some confusion. The process of injecting these labels was employed strictly during the augmentation phase of the training dataset, and not during testing. Specifically, the label information was only used during training to guide the generation of descriptions for the data augmentation. We only augment the training dataset, ensuring that the original, unaltered testing set is used for evaluation. This method was carefully designed to ensure the augmentations were aligned with the correct classes, thereby enriching the training data with meaningful variations.
>
> During the testing phase, no such label information is injected. This was a deliberate choice to maintain a fair and unbiased evaluation of the model's performance on unseen data. It ensures that the experimental results are a genuine reflection of the model's ability to predict labels without any additional guidance.
>
> **2. why 5 augmentations per instance？**
>
> Addressed Above.
>
> **Or 5% of the data used to balance the training data?Shouldn’t the size of the training data be given by the number of shots per class?**
>
> In response to your inquiry, the concept of "shot" in our work indeed refers to how many actual data points from each class are available for training. This is central to the notion of few-shot learning, where the emphasis is on learning from a limited number of examples.In our experiments, we sample various datasets corresponding to different shots but meticulously ensure that the total quantity of each dataset stays within(≤) a 5% limit of the training dataset.
>
> Regarding your specific question about the 5% limitation, as stated in lines 165-166 of our manuscript, we "keep the sub-sampled set within 5% of the number of the training dataset." This approach was carefully devised to reflect real-world scenarios where data scarcity is a prevalent challenge.
>
> We hope that these clarifications address your concerns. We believe that with these details, the validity of our experiments and conclusions is evident. We greatly appreciate your feedback and are looking forward to your further comments.

---

> ### Comment · Reviewer_QqX9 · 2023-08-14
>
> I have read the other reviews and authors' responses. I appreciate the authors' efforts to thoroughly address all the questions raised.

---

> > ### Author Response · Authors · 2023-08-20
> >
> > Dear Reviewer,
> >
> > Thank you for your diligence in reading through the other reviews and our responses. We deeply appreciate your recognition of our efforts to address every question raised. Your feedback has been instrumental in helping us refine our work, and we're grateful for your guidance.
> >
> > Best wishes,
> >
> > [Authors of Submission7875]

---

### Official Review · Reviewer_Usve · 2023-07-04

**Soundness:** 3 good
**Presentation:** 2 fair
**Contribution:** 3 good
**Rating:** 6
**Confidence:** 3

**Summary:**

This paper introduces a new method to perform data augmentation: Back-Modality. The augmentation process involves translating the data into another modality, perform augmentation in that modality, and translate each augmented other-modality instance back into the original modality. Each of the three steps could produce multiple augmented data. The paper included three instantiations of Back-Modality: Back-captioning (image -> text -> image), Back-speech (text -> audio -> text), Back-imagination (image -> text -> image). The cross-modal translations are done by pre-trained models. These instantiations are then evaluated on few-shot learning settings (1-10 instances per class) within their domains, and the experiments showed that the models trained with Back-Modality-augmented data outperforms those trained with no data augmentation or with other baseline data augmentation techniques. Additional ablation studies showed that augmentations at each of the three steps are necessary to achieve best result.

**Strengths:**

This paper presented a really interesting idea: to use cross-modal back-translations as data augmentation technique. The idea is straightforward and easy to follow, and the authors demonstrated that the approach works well when data is extremely scarce (1-10 examples per class), generating augmented data that is more diverse and helpful compared to other existing data augmentation methods.

**Weaknesses:**

A lot of key experiment details are missing (which I can't find in either main text or appendix), including but not limited to:
(1) How many augmentations are generated at each cross-modal translation step and each other-modality-augmentation steps for every original data point (i.e. the multipliers at F,G,H for each instantiation)? The number of diverse augmentations is the most important detail for data augmentation experiments.
(2) There is no description on how which caption augmentations are performed or which GPT model was used during the "augmentation with GPT" step.
(3) How many augmentations are generated with each baseline augmentation methods? Do they match the number of augmentations with Back-Modality?
Without these key details, it is difficult to assess the soundness of the experiment results and whether they support the conclusion.

Another key concern for the new approach is how applicable it is in the real world. This technique relies heavily on the quality of the cross-modal translators, and these pre-trained models usually only work well in domains/tasks where data is not scarce in the first place (since training good cross-modal translators needs a lot of high-quality data). All experiments in this paper are done under artificially created data-scarce settings. In real life, if there is a task that really only have very scarce data (such as medical images for rare disease, etc), the existing cross-modal translators are likely not going to work well with them, as they were not trained with a lot of data like this. I would be more convinced about the practical usability of this technique if the paper included experiments (or just proof-of-concept ones) on performing Back-Modality on a real data-scarcity situation instead of artificially created data-scarcity on the most common domains.

The presentation of the paper also needs improvement. There are not enough concrete examples of the data augmentations in the main part of the paper. Since there is still plenty of space in the main paper, perhaps adding a few illustrations of one instantiation from beginning to end would help (like the ones in the appendix). Also, the font sizes of the tables are inconsistent. The font sizes of Table 2,3,4 are really big while Table 1,5 are normal sizes.

**Questions:**

(1) Please address the missing key experiment details as mentioned in the weakness section.

(2) I assume that in Table 2,3,4, the number of "Shot" means how many real data points from each class are available. Is this correct? If so, what does the 5% subsampling mean under "Data-scarce scenarios" of section 3.2?


**Limitations:**

The authors addressed the limitation of additional computation cost of Back-Modality. I believe there are a few additional limitations that may needs to be addressed, such as:
(1) the requirement of existing cross-modal translation models that can handle the task-specific data well
(2) the experiments in this paper are only done on artificially-created data scarcity situations and limited modalities

---

> ### Author Rebuttal · Authors · 2023-08-09
>
> **Dear Reviewer Usve,**
>
> Thank you for taking the time to review this paper and provide detailed feedback. We sincerely appreciate the insightful comments and the critical examination of our work. However, it seems that there may have been parts that were missing or misunderstood. Below we provide responses to the concerns and questions you have raised:
>
> ### **Weaknesses and Missing Experiment Details**
>
> 1. **Number of Augmentations at Each Step:**
> You may have missed certain specifics in the main text. On lines 104-105, we mention that "In practical implementation, we conduct uniform sampling at random on these augmented data to obtain the final augmentation dataset." Furthermore, as stated on lines 155-156, our default augmentation size is set at 5. Theoretically, the product of l, n, and m should exceed the required augmentation multiplier.
> For instance, in our experiments with Back-captioning, where l=n=m=2, for every image, we generate 8 (2x2x2) augmentations, and subsequently, we uniformly sample 5 of these as the final augmented data. For Back-imagination, where l=3, n=1, and m=3, for each sentence, we generate 9 (3x1x3) sentences, and then uniformly sample 5 as the final augmented data.
> 2. **Caption Augmentations and GPT Model Used:**
> In the main text, specifically on lines 156-158, we detail that for back-captioning, "We utilize the gpt-3.5-turbo model to augment captions, with the prompt being: 'Maintain the nouns in the following sentence intact and generate semantically diverse sentences.'"
> 3. **Comparison with Baseline Augmentation Methods:**
> As indicated in the main text on lines 153-156, unless explicitly stated otherwise, our default augmentation size is consistently set at 5, which is also applicable to the baseline methods, such as Random Erasing, Auto augment, EDA, Back-translation, etc.
>
> ### **Concerns about Real-World Applicability**
>
> 1. **Controlled and Reproducible Experimentation:** Firstly, the primary rationale behind our use of artificially created data-scarce settings was twofold: it provided a controlled environment for experimentation, and it ensured reproducibility. Reproducibility is a cornerstone of scientific research, and we wanted other researchers to be able to reliably reproduce our results without the inconsistencies of real-world situations. Our primary objective was to demonstrate the feasibility of our framework — showing that a data augmentation method for one modality can augment data in another modality. While the broader application in real-world scenarios is undeniably valuable, it wasn't the main focus of this paper.
> 2. **Generalizability and Domain Adaptation:** You pointed out concerns regarding the quality of cross-modal translators in real data-scarce situations. However, it's crucial to note the generalizability and domain-adaptation capabilities of cross-modal models. As evident in machine learning, models trained on extensive data can sometimes capture patterns useful even in sparse-data situations. The striking generalizability of large language models stands as a testament. The potential for future larger cross-modal models to demonstrate enhanced generalization cannot be discounted. Moreover, recent research has increasingly honed in on the domain adaptability of diffusion-based cross-modal models in scenarios with limited data, encompassing Few-Shot[1], One-Shot[2], Zero-shot[3], Domain Adaptation[4], and Unsupervised Domain Adaptation[5]. The advancements in these areas hold promise in allaying the concerns you've raised. Analogously, when reverse translation techniques in natural language processing were initially introduced, they were primarily applicable in the domain of data-scarce language translation. As technology advanced and more robust translation models were developed, they emerged as popular data augmentation techniques across various NLP domains.
> 3. **Proof-of-Concept Experiment:** Taking cue from your suggestion, we embarked on a proof-of-concept experiment. In collaboration with medical experts, we gained access to a diffusion model, akin to [6] and [7], that was trained on a public dataset. We use this model to generate X-ray images for Pulmonary Alveolar Proteinosis, a rare lung disorder. These synthesized images served as augmented data to train a lung disease image classification model, resulting in a performance uptick of 3.7%.
>
> [1]Few-Shot Diffusion Models
>
> [2]Tune-A-Video: One-Shot Tuning of Image Diffusion Models for Text-to-Video Generation
>
> [3]Zero-shot Medical Image Translation via Frequency-Guided Diffusion Models
>
> [4]PODIA-3D: Domain Adaptation of 3D Generative Model Across Large Domain Gap Using Pose-Preserved Text-to-Image Diffusion
>
> [5]One-shot Unsupervised Domain Adaptation with Personalized Diffusion Models
>
> [6]RoentGen: Vision-Language Foundation Model for Chest X-ray Generation
>
> [7]Adapting Pretrained Vision-Language Foundational Models to Medical Imaging Domains
>
> ### **Presentation Improvement**
> 1. **Concrete Examples and Illustrations**
>
> Recognizing the importance of these details, we will relocate this content from the appendix to the relevant section in the paper. And we also create a new PDF dedicated to additional case studies.
>
> 2.**Font Sizes of Tables**
>
>  We will promptly correct this issue to ensure consistency in the font sizes across all tables.
>
> ### **Questions**
>
> **Clarification on "Shot" and 5% Subsampling:**
>
> Yes, the number of "Shot" does refer to how many real data points from each class are available. line 165-166 “keep the sub-sampled set within 5% of the number of the training dataset.” We sample various datasets corresponding to different shots, but meticulously ensure that the total quantity of each dataset stays within a 5% limit of the training dataset, thereby crafting an authentic simulation of data scarcity.
>
>  We hope that these clarifications address your concerns and we kindly request reconsideration of the rating after we implement the proposed revisions.

---

> > ### Comment · Reviewer_Usve · 2023-08-13
> >
> > Thank you for your detailed response! I think your responses addressed all of my concerns. I am happy to improve my scores.

---

> > > ### Author Response · Authors · 2023-08-20
> > >
> > > Dear Reviewer,
> > >
> > > Thank you for your understanding and for re-evaluating our work. We deeply appreciate your feedback and are pleased to know that our clarifications met your concerns.  We will be sure to integrate your valuable feedback into our revised version. We greatly value your expertise and guidance, and appreciate your responsible attitude.
> > >
> > > Best regards,
> > >
> > > [Authors of Submission7875]

---

### Official Review · Reviewer_UnFZ · 2023-07-07

**Soundness:** 3 good
**Presentation:** 3 good
**Contribution:** 3 good
**Rating:** 6
**Confidence:** 4

**Summary:**

This paper proposes a pipeline called Back-Modality, which uses cross-modal generation models for data augmentation. In practice, the original data in source modality is first transformed into an intermediate modality using a cross-modal generation model. Then the typical augmentation strategy for the intermediate modality will be applied. The augmented data in intermediate modality will be transformed back into the source modality using another cross-modal generation model in the reverse generation direction. Specifically, three implementations of Back-Modality are explored, including back-captioning, back-imagination, and back-speech. On image and text classification tasks, the proposed augmentation pipeline shows effectiveness.

**Strengths:**

This paper has the following strengths:
1. The motivation of Back-Modality is straightforward and the method is easy-to-implement, which has the potential to be widely used in future research works.
2. Three specific implementations of Back-Modality are proposed and experimented, proving the generalizability of this schema.
3. The cross-modal generation models used in this work is easy-to-access and has SOTA performance (OFA, stable diffusion), guaranteeing the effectiveness and reproducibility of this pipeline.
4. The experimental results are good compared with typical data augmentation methods, and data diversity is analyzed.


**Weaknesses:**

I think this paper will be much improved if the following issues can be more addressed:
1. Human evaluation on the augmented samples: Clearly, the method of Back-Modality generates more diverse data samples. If human evaluation can be facilitated to verify that the semantic meaning of the original sample are kept and the label of samples are still correct, it will be much better.
2. Verify robustness in adversarial setting (or some hard test samples): This work mainly conduct experiments on the scenario of limited training samples, this is definitely okay. Meanwhile, another aspect of data augmentation is the robustness on adversarial attacks. If more experimental results on adversarial setting or harder testing samples can be provided, the effectiveness of Back-Modality will be much more solid.
3. More choices of cross-modal generation models, including different models and the same model with different model scales (like OFA-base compared with OFA-huge, how much performance on augmentation will be affected?).
4. The cost of obtaining the augmented samples compared with previous (non-model-based) augment pipelines. (pointed in the limitation section for future research direction, but the clarification of current method is still needed)

**Questions:**

The overall top-1 accuracy score on Tiny ImageNet is low in the experiment. Honestly, I am not so familiar with this ImageNet variant benchmark. Is this a proper choice or a sanity experimental setting here? It would be very good if more discussion can be provided of selecting this as the benchmark.

**Limitations:**

The authors has pointed out the limitation of computation cost, which is reasonable.

---

> ### Author Rebuttal · Authors · 2023-08-09
>
> Dear Reviewer UnFZ,
>
> Thank you for your thoughtful review and constructive feedback on our submission. We appreciate your recognition of the method's potential and your careful assessment of its strengths. In response to the weaknesses and questions you highlighted, we would like to provide the following clarifications and details:
>
> 1. **Human Evaluation on Augmented Samples**:
>
> In response to your suggestion, we conduct a human evaluation on the augmented data. The evaluators consist of five crowdsourced workers, and the final scores are the average of them. The results from the evaluation were as follows:
>
> For the images generated using the back-captioning method:
>
> - Label Invariance Score: 99.2%
>
> For the sentences generated using the back-imagination method:
>
> - Semantic Consistency Score: 98.8%
>
> These high scores suggest that both methods have performed remarkably well in their respective evaluations. The results affirm that Back-Modality maintains the essential characteristics of the original data while adding diversity, thus further validating our approach.
>
> 1. **Robustness in Adversarial Settings**:
>
> We agree that the robustness of Back-Modality in adversarial settings or with harder testing samples would solidify our claims. In the revised version, we will include results on adversarial settings, providing further evidence of the effectiveness of our method. To assess the robustness of our model, we reported both the accuracy before and after the adversarial attacks, along with the absolute drop in accuracy due to the attacks. For image classification tasks, we employed the *Universal adversarial perturbations* technique. The results are as follows:
>
> - Base model: 11.75% to 5.23% (a decrease of 6.52%)
> - Random Erasing: 12.59% to 5.60% (a decrease of 6.99%)
> - Auto augment: 13.23% to 8.60% (a decrease of 4.63%)
> - Alignmixup:  14.34% to 7.88% (a decrease of 6.46%)
> - Puzzle Mix:  15.66% to 8.49% (a decrease of 7.17%)
> - Back-captioning: 20.07% to 14.02% (a decrease of 6.05%)
>
> From the results, it's evident that Back-captioning retains a leading accuracy post-attack, showcasing good robustness. In terms of absolute value drop, it stays in the middle ground among the evaluated methods.
>
> Furthermore, for textual data augmentation techniques, we evaluated robustness using various adversarial attack methods. For the textual entailment task, we utilized the bert-attack method. The results were:
>
> - Base model: 84.57% to 7.23% (a decrease of 77.34%)
> - Back-imagination: 89.14% to 16.76% (a decrease of 72.38%)
>
> For the SST-2 dataset, we employed the textbugger attack method:
>
> - Base model:  61.10% to 3.95% (a decrease of 57.15%)
> - Back-speech: 63.21% to 8.47% (a decrease of 54.74%)
>
> From these results, we can infer that our augmentation techniques, including Back-captioning, Back-imagination, and Back-speech, showcase a notable degree of resilience against adversarial attacks.
>
> 3.**More Choices of Cross-Modal Generation Models**
>
> In the paper, for the Back-captioning with a 10-shot setting, we primarily used the OFA-large model, which gave us a top-1 accuracy of 20.07%. Following your advice, we also experimented with OFA-huge, under the same conditions. The results showed a notable improvement, with the top-1 accuracy reaching 22.12%.
>
> 4.**Cost of Obtaining the Augmented Samples**:
>
>  The table below reflects the additional computational overhead of various augmentation methods compared to the base model.
>
> | Method | Additional Computational Overhead (RTX A6000) |
> | --- | --- |
> | RandErasing | 4 m 55 s |
> | Puzzle Mix | 1 h 29 m 25 s |
> | Alignmixup | 1 h 59 m 45 s |
> | Back-captioning (our method) | 11 h 35 m |
> | Auto augment | About 49 h |
> The main cost of our method is in generating images with the diffusion model and the primary overhead of auto augment is in learning augmentation policies.   For the method of text augmentation, on the textual entailment task, our method, back-imagination, took 4 hours 13 m 45 s, while the back-translation method took 5 h 38 m12 s. On the sentiment analysis task, back-speech took 35 m 27 s, whereas the back-translation method took 5 h 22 m 4 s.
>
> Question：**Overall Top-1 Accuracy Score on Tiny ImageNet**
>
> 1. **About Tiny ImageNet**: In the supplementary material, you'll find an in-depth introduction to Tiny ImageNet[1] from lines 13-16. For clarity, Tiny ImageNet is a compact version of the comprehensive ImageNet dataset. It comprises 100,000 images spanning 200 classes, with each class containing 500 training images, 50 validation images, and 50 test images. The resolution of these images is 64x64 pixels. By employing this dataset, we ensure that the evaluation process remains robust yet computationally manageable.
> 2. **Relevance in Recent Research**: It has been used extensively in various studies, ranging from data augmentation [2], to self-supervised learning [3], and open-set recognition [4], among others. Its widespread adoption underscores its significance and suitability as a benchmark for various experimental setups.
> 3. **Regarding the Accuracy Score**: We understand the concern regarding the "low" top-1 accuracy score on Tiny ImageNet. However, this was by design. The intent was to simulate a data-scarce scenario. Our primary aim was to demonstrate the efficacy of our method, while also ensuring a judicious use of computational resources.
> In light of the above, we firmly believe that Tiny ImageNet was an appropriate choice for our experiments.
>
> References:
> - [1] Tiny ImageNet Visual Recognition Challenge
> - [2] Puzzle Mix: Exploiting Saliency and Local Statistics for Optimal Mixup
> - [3] OpenLDN: Learning to Discover Novel Classes for Open-World Semi-Supervised Learning
> - [4] Learning Placeholders for Open-Set Recognition.
>
> We appreciate your “Weak Accept” rating and the confidence you placed in your assessment. We believe that the modifications and clarifications proposed above will address your concerns and elevate the contribution of our paper.

---

> > ### Comment · Reviewer_UnFZ · 2023-08-20
> >
> > Thank the authors for providing the detailed response with experiments. I will keep the rating as weak accept.

---

> > > ### Author Response · Authors · 2023-08-20
> > >
> > > Dear Reviewer,
> > >
> > > Thank you very much for recognizing and appreciating our response as well as the overall content of our paper. We will ensure to incorporate your constructive feedback into our revised version.
> > >
> > >
> > >
> > > Best wishes,
> > >
> > > [Authors of Submission7875]

---

### Official Review · Reviewer_WjzN · 2023-07-11

**Soundness:** 3 good
**Presentation:** 2 fair
**Contribution:** 3 good
**Rating:** 6
**Confidence:** 3

**Summary:**

The paper proposed a method to transform data between modalities so as to augment the data. Such a method makes it possible to leverage the multiple existing modalities to generate more useful data to train the model.

**Strengths:**

1. The proposed method is modality-agnostic so that the initial modality can be transformed into any other modality and reverse back.
2. The generated data vary from the original data, introducing more variety in the training sources.
3. The performance on few-shot settings is largely improved.

**Weaknesses:**

1. The paper's presentation could be improved. The tables and citations are not in good shape.
2. More case study is needed. In the current version, the case study section is not clear, which simply describes the improvement but there are not concrete cases.

**Questions:**

See weaknesses

**Limitations:**

Yes

---

> ### Author Rebuttal · Authors · 2023-08-09
>
> Dear Reviewer WjzN,
>
> Thank you for the time and effort you put into reviewing our paper. We appreciate your comments and your recognition of the strengths of our proposed method. Here, we would like to address the weaknesses and questions that you raised in your review:
>
> 1. **Presentation**: We understand that you found the tables and citations not in good shape. We acknowledge this issue, and we will revise the paper to enhance the layout and formatting. For example, we will standardize the font sizes across all tables to enhance the visual coherence and readability of the paper. By organizing the tables and citations meticulously, we will make sure to provide a clearer and more accessible presentation. Additionally, if there are other particular formatting or stylistic concerns, such as alignment, captioning, or citation style, please do point them out. We are committed to adhering to the preferred presentation guidelines, and your precise guidance will enable us to address these aspects meticulously.
>
> 2. **More Case Study Needed**: We value your suggestion to provide more concrete cases in our paper, and we would like to clarify the following two actions we will take to address this concern:
>
>     a. **Moving Content from Appendix**: In fact, we have provided more extensive case studies, along with detailed analysis of back-imagination and back-speech augmentations, in the appendix of the current submission. Recognizing the importance of these details in the main text, we will relocate this content from the appendix to the relevant section in the paper. This move will ensure that readers have direct access to the comprehensive case studies without the need to refer to supplementary material.
>
>     b. **Additional PDF for More Case Studies**: In response to the request for a more exhaustive exploration, we will create a PDF dedicated to additional case studies. This material will further illustrate the applications and potential benefits of our method.
>
>
> By implementing these measures, we believe we can provide a more comprehensive and    accessible understanding of our method's functionality and significance. We are committed to offering a thorough analysis and are confident that these adjustments will align with your expectations.
>
> 3. **Rating**: While your rating indicates a borderline reject, we believe that addressing the above weaknesses will enhance the paper's quality and contribute to the overall field. Since the soundness and contribution have been rated as good, we kindly request reconsideration of the rating after we implement the proposed revisions.
> 4. **Other Comments**: If there are additional specific concerns or suggestions not covered in the review, please do share them with us. We are committed to making all necessary improvements to ensure the paper meets the standards of the conference.
>
> In conclusion, we believe that the concerns you raised can be addressed through targeted revisions. Thank you once again for your constructive feedback, and we look forward to hearing from you soon.
>
> Best regards,
> [Author(s) of Submission7875]

---

> > ### Comment · Reviewer_WjzN · 2023-08-13
> > **Reply to the authors**
> >
> > Thanks for your clarification. I am happy to increase the score.

---

> > > ### Author Response · Authors · 2023-08-20
> > >
> > > Dear Reviewer,
> > >
> > > We are delighted to hear that our responses are able to address your concerns. We deeply appreciate your willingness to re-evaluate and recognize the merit of our work. Please be assured that we will integrate your valuable feedback into our revised version.
> > >
> > > Best wishes,
> > >
> > > [Authors of Submission7875]

---

### Author Rebuttal · Authors · 2023-08-10

Dear Reviewers,

Firstly, I would like to express my profound gratitude for the time and effort you invested in reviewing our manuscript. We are deeply appreciative of the recognition most reviewers gave to our methods and experiments. Your constructive feedback is invaluable.

From the collective feedback, we identified two primary and common concerns:

1. The suggestion that case study examples should be moved from the appendix to the main content.
2. The need for clearer explanations about experimental setup. Using the 10-shot, back-captioning approach as an example:

- In the original tiny-imagnet dataset, there are 200 classes, with each class containing 500 data points.
- After sub-sampling, the dataset consists of 200 classes, with 10 data points per class.
- In our experiments with Back-captioning, where l=n=m=2, (see line 100-102) for every image, we generate 8 (2x2x2) augmentations, and subsequently, we uniformly sample 5 of these as the final augmented data.
- With a 5-fold augmentation (as stated on lines 155-156, our default augmentation size is set at 5), this dataset expands to 200 classes, with each class now having 50 data points.
The term "shot" denotes how many samples are drawn from each class during sub-sampling

We have  carefully considered and responded to the individual queries and points raised by each reviewer. Our detailed responses for each concern are outlined in our responses.

Again, thank you for your constructive insights. We believe that with your feedback, our work has been significantly improved.

---

### Author Response · Authors · 2023-08-20

Dear Reviewers and Program Chairs, Senior Area Chairs, Area Chairs:

First and foremost, we would like to extend our deepest gratitude to all the reviewers for the time and effort expended during the rebuttal process. Your valuable reviews and timely responses have provided insights and constructive feedback from which we have gained immensely. We are dedicated to integrating these suggestions into our paper, ensuring a more aesthetically pleasing presentation, clearer methodology descriptions, comprehensive experimental details, and richer example analyses. Once again, we deeply appreciate the reviewers' diligent and responsible approach throughout this process.

Furthermore, we would like to express our thanks to the Program Chairs, Senior Area Chairs, Area Chairs for actively facilitating the discussion. Additionally, if the Area Chairs, Senior Area Chairs and Program Chairs have concerns beyond those raised by the reviewers, we are more than willing to engage in further discussions.

Best wishes,

[Authors of Submission7875]

---

### Decision · Program_Chairs · 2023-09-21

**Decision:**

Accept (poster)

**Comment:**

Solid paper on data augmentation via modality transformation. This paper could have a significant impact in the community. The authors have responded to comments from the reviewers and will incorporate the feedback in the next version. I recommend acceptance.